# A Phylogenetic Perspective on the Evolutionary Patterns of the Animal Interleukin-10 Signaling System

**DOI:** 10.3390/genes16111243

**Published:** 2025-10-22

**Authors:** Liu Tang, Zeyu Zhou, Weibin Wang, Dawei Li, Tingting Hao, Yue Chen

**Affiliations:** 1State Key Laboratory for Conservation and Utilization of Bio-Resources in Yunnan, Yunnan Agricultural University, Kunming 650201, China; tangliu@dongyang-lab.org (L.T.); zhouzy@dongyang-lab.org (Z.Z.); 2College of Food Science and Technology, Yunnan Agricultural University, Kunming 650201, China; 3College of Science, Yunnan Agricultural University, Kunming 650201, China; wangwb@dongyang-lab.org; 4College of Plant Protection, Yunnan Agricultural University, Kunming 650201, China; dli@dongyang-lab.org

**Keywords:** interleukins, phylogenetic tree, evolution of interleukins

## Abstract

**Background:** The interleukin-10 (IL-10) signaling system, comprising ligands (IL-10s) and receptors (IL-10Rs), plays critical roles in immune regulation, inflammation resolution, and disease pathogenesis. “IL-10 signaling system” here refers to the immunomodulatory signaling system composed of ligands (IL-10s) and receptors (IL-10Rs), which belong to different Protein families in evolution, but achieve functional synergy through the conserved JAK-STAT pathway. Understanding their evolutionary and functional dynamics is essential for elucidating immune mechanisms and therapeutic targeting. **Methods:** Through phylogenetic reconstruction, homology analysis, and sequence alignment across >400 animal species, we traced the evolutionary trajectory and structural–functional diversification of IL-10s and IL-10Rs. **Results and Conclusions:** IL-10 signaling components emerged in early vertebrates, with IL-10Rs originating in cartilaginous fishes (~450 Mya) and IL-10s diversifying in bony fishes (~400 Mya). Functional divergence yielded immunosuppressive (IL-10), barrier-protective (IL-20 subfamily), and antiviral (type III IFN) subgroups. Structurally, conserved motifs (e.g., IL-10R1 GYXXQ, IL-22 N54-glycosylation) underpin receptor–ligand binding and JAK/STAT signaling. Evolutionarily invariant residues suggest candidate therapeutic epitopes. This study provides an evolutionary framework highlighting functional conservation and species-specific adaptation within IL-10 signaling, with implications for immunotherapy and animal breeding.

## 1. Introduction

Cytokine Interleukin-10 (IL-10) is an important regulator of immune cell function, proliferation, and survival [1]. Herein, the term ‘IL-10 signaling system’ refers to the functional network comprising IL-10 cytokines (including IL-10, IL-20, IL-22, IL-28 subfamilies) and their cognate receptors (IL-10R1, IL-10R2, IL-20R1/2, IL-22R1/2), which transduce signals via the conserved JAK-STAT pathway. To avoid confusion with protein families, we distinguish between the ‘IL-10 cytokine subfamily’ (ligands) and the ‘IL-10 receptor subfamily’ (receptors). The IL-10 cytokine subfamily can be divided into three subgroups according to its function: IL-10, IL-20 subfamily, and type III interferon. IL-10 mainly targets innate and adaptive immune responses and has immunosuppression effects to reduce excessive inflammatory reaction [2]. The IL-20 subfamily, including IL-19, IL-20, IL-22, IL-24, and IL-26, mainly acts on epithelium cells and stromal cells, inducing innate host defense mechanisms, protecting barrier integrity, and promoting tissue repair [2]. Type III interferons are distantly related cytokines to IL-10, including IL-28A (IFN-λ2), IL-28B (IFN-λ3), and IL-29 (IFN-λ1) [3,4,5,6], mainly targeting epithelium cells, and have overlapping biological functions and downstream signaling pathways with type I interferons [2]. The IL-10 receptor subfamily belongs to the type II cytokine receptor family and are mainly composed of two types of receptor subunits. Type R1 receptors have long intracellular domains, including IL-10R1, IL-20R1, and IL-22R1 (IL-22R), etc. Type R2 receptors have short intracellular domains, mainly including IL-10R2 and IL-20R2 [7,8,9]. These receptor subunits can form different heterodimeric complexes for recognizing and binding different IL-10 cytokine subfamilies [10,11]. IL-10 cytokine subfamily proteins/subunits are represented in this article by IL-10s, whereas IL-10Rs represent IL-10 receptor subfamily proteins/subunits. The IL-10 system, composed of IL-10s and IL-10Rs, is illustrated in Figure 1 [8,11,12]. IL-10 constitutes an intercalated dimer, with each monomer adopting six-helix bundle domains, while most of the other IL-10 cytokine subfamily members, such as IL-19 and IL-22, are monomers composed of seven amphipathic helices A-G of varying lengths, forming a seven-helix bundle with a compact hydrophobic core. They all belong to the class II α-helix cytokine family. The predicted helical structures of these cytokines are conserved, but certain receptor-binding residues are variable, which determines their interaction with different types of class II cytokine receptor-specific heterodimers [13,14]. IL-10 cytokine subfamily members have limited homology in primary sequence but significant structural similarity [10].

Differences exist among the IL-10 cytokine subfamily in terms of chromosome localization and signal transduction processes. The genes encoding IL-10, IL-19, IL-20, and IL-24 are located on human chromosome 1q32; those for IL-22 and IL-26 are located on human chromosome 12q15; the genes encoding IL-28A, IL-28B, and IL-29 are located on human chromosome 19q13 [15], indicating that they may have a common ancestor and may have separated during evolution to form 3 gene clusters. After binding to their corresponding ligands, the IL-10 receptor subfamily induces conformational changes and activates downstream Jak/STAT signaling pathways, especially the activation of STAT3 and STAT1. This signal transduction can rapidly activate multiple transcription factors, thereby regulating gene expression [8,16]. IL-10 cytokine subfamily members form heterodimeric complexes in different combinations by binding to shared class II cytokine receptor chains [16] (Figure 1). For example, IL-10R2 is a receptor subunit shared by multiple IL-10 cytokine subfamily members [17]. IL-10 binds to IL-10R1/IL-10R2 and leads to phosphorylation of STAT3, a key transcription factor for the immunosuppression effect of IL-10 [18]. IL-19, IL-20, and IL-24 induce their signaling through the IL-20R1/IL-20R2 receptor complex. IL-22 and IL-24 can both induce their signaling through the IL-22R/IL-20R2 receptor complex. IL-22 and IL-26 function through receptor complexes composed of IL-22R/IL-10R2 and IL-20R1/IL-10R2, respectively. IL-28A, IL-28B, and IL-29 bind to the receptor complex formed by IL-28R/IL-10R2 [19] (see Table 1).

Within the IL-10 signaling system, the genes encoding each member are closely clustered, their genomic structures are conserved, and they share common features in their primary and secondary protein structures, as well as in their receptor-binding modes [4]. Despite structural similarities, IL-10 signaling system cytokines exert different functions in different species; for example, IL-22 has anti-bacterial effects in murine animals, whereas in humans it is associated with immunomodulation [2]. The high polymorphism of the IL-10 signaling system may be due to the diversity of expressing cells (see Table 1). In addition, interleukin IL-10 signaling system-mediated biological responses also include immunosuppression, enhanced anti-bacterial and antiviral capabilities, increased anti-tumor activity, and promotion of self-tolerance in autoimmune disorders (see Table 1).

IL-10 signaling system members play multiple roles in the immune system. As a key immunosuppressive cytokine, IL-10 acts on leukocytes to resolve infections and inflammation, with functions including reducing tissue injury, inhibiting antigen-presenting cell function, and promoting the survival and function of regulatory T cells [2]. For example, mast cells (MCs) can produce IL-10 under specific stimulation (such as IgE cross-linking, IL-33, or pathogen infection). The produced IL-10 exerts a dual role through autocrine or paracrine pathways: it suppresses excessive immune response, enhances Treg function, and maintains tissue homeostasis in chronic inflammation (such as GVHD, urinary bladder immune privilege), while promoting MC activation and inflammatory mediator release in acute anaphyl (such as food allergy). This highlights that its role is highly dependent on the microenvironment, time scale, and disease type, making it a key regulator of dynamic balance in the immune network [27]. The IL-28 subgroup, classified as a type III interferon, mainly targets epithelial cells and has overlapping biological functions and downstream signaling pathways with type I interferons [2]. The cytokines, IL-28A, IL-28B, and IL-29, all exhibit antiviral activity [28]. The IL-20 subgroup promotes the proliferation and repair of epithelial cells and stromal cells, and protects barrier integrity [2]. IL-20 subfamily members primarily target the non-immune compartment, such as tissue epithelial cells, to stimulate innate defense mechanisms to control viruses, bacteria, and fungal infection, protect tissue integrity, and promote tissue repair and regeneration [3,29]. For example, IL-22 directly acts on epithelial cells, enhances host defense, and promotes tissue repair, playing an anti-bacterial and anti-fungal role in tissues, such as the intestine [2]. The IL-10 receptor complex plays a key role in inhibiting inflammatory reactions and regulating immune homeostasis. It can inhibit the activation and effector functions of T cells, monocytes, and macrophages, limit and terminate inflammatory reactions, and also participate in regulating the growth and differentiation of B cells, natural killer cells, cytotoxic and helper T cells, mast cells, Granulocytes, dendritic cells, keratinocytes, and endothelial cells [12,30]. The IL-10 receptor subfamily is vital in maintaining the integrity and homeostasis of the tissue epithelium layer, which can promote innate immune response from the tissue epithelium, limit injury caused by virus and bacterial infection, and facilitate tissue healing after infection or inflammation [5,29].

Although each member has specific functions, they share commonalities in regulating immune responses, maintaining tissue homeostasis, and participating in host defense [5,31]. IL-10 signaling system members play a crucial role in immune regulation and host defense by suppressing excessive inflammation, promoting tissue repair, and enhancing host defense, helping the body to resist pathogen invasion and maintain tissue homeostasis [2]. Potent anti-inflammatory and immunomodulatory properties of IL-10 bring broad prospects among clinical application, with clinical trials in areas such as rheumatoid arthritis, inflammatory bowel disease, tumor immunotherapy, and chronic viral infection. Despite challenges like side effects and tolerance of systemic administration, the medical potential of IL-10 remains promising and merits further investigation [3]. IL-10 signaling system members are crucial in the immune system, and their evolutionary relationships and functional differences are of great significance for revealing the evolutionary mechanism of the immune system. Studying the evolutionary relationships of the IL-10 signaling system helps to deeply understand the process and laws of immune system evolution.

From ancient fish to mammals, IL-10 signaling system genes exhibit high conservation during evolution, while also demonstrating species-specific functional adaptation [32,33,34,35,36,37,38]. Functionally, IL-10 signaling system members exhibit diversity, with IL-10 mainly playing an anti-inflammation role, while members such as IL-19, IL-20, IL-22, IL-24, and IL-26 contribute significantly in regulating skin inflammation, wound healing, and intestinal immunization [39]. By comparing IL-10 signaling system members from different species, clear evolutionary relationships can be found between them. For example, IL-10 signaling system members in humans and mice also exhibit certain differences in gene regulation. For example, the promoter region of IL-10 contains multiple transcription factor binding sites, while the regulatory mechanisms of other family members may be more complex. Family members may have functional synergy and jointly regulate immune response. For example, IL-22 and IL-10 can jointly regulate intestinal immunity. The evolutionary relationship of the IL-10 signaling system shows a clear affinity, while family members show certain differences in function [39]. Because IL-10 signaling system have presented potential application value in the diagnosis of multimorbidity [40,41,42], showing bright prospects of application in animal breeding and productivity improvement [36,37,43], a comparison can be carried out between large-scale cross-species IL-10 signaling system, as well as phylogenetic analysis, which not only helps understand the evolutionary history of IL-10 signaling system, but also provides important clues for exploring its functions under different physiological and pathological conditions. This can provide a theoretical basis for developing targeted therapeutic strategies for the IL-10 signaling system of different animals.

In summary, the IL-10 signaling system of cytokines exhibits high structural similarity, receptor usage, and conservation in evolution, but with some functional differentiation. Such functional differentiation enables the IL-10 signaling system of cytokines to play different biological roles among different species, providing a major reference for utilizing these cytokines in host defense and disease treatment. The IL-10 signaling system of cytokines is also essential in the immune system, including regulating immune response, controlling inflammation, and promoting tissue repair. However, despite significant progress in functional regulation, there is still a lack of systematic research on how these cytokines evolved and their specific roles in host defense. Therefore, this study investigates the distribution, evolutionary patterns, and sequence traits of IL-10s and IL-10Rs in over 400 species by utilizing comparative genomics and phylogenetic analysis and analyzes their conservation and evolutionary relationships to reveal the evolutionary history and functional roles of these cytokines in host defense. By exploring the structural characteristics and functional adaptation of these genes in different species, it reveals that IL-10 signaling system genes are highly conserved in evolution, but they also exhibit species-specific functional differences, which are of great significance for understanding the evolution of the immune system, and the diagnosis and treatment of related diseases. This research background helps to further understand the biological significance of the IL-10 signaling system cytokines and provides theoretical support for their clinical applications in the future.

## 2. Materials and Methods

### 2.1. Construction of a Phylogenetic Tree

We obtained the genomes of 520 individual animals along with their respective GFF annotation files. The genome data primarily originated from various databases, including NCBI (248 individuals), Ensembl (265 individuals), CNCB (6 individuals), and Macgenome (1 individual). Using the mammalia_odb10 database, we utilized the GPA script (https://github.com/ypchan/GPA, accessed on 1 October 2025) to assess genome assembly quality and extract statistical information from BUSCO ortholog (protein) sequences. We retained species that met the criteria of C > 2.4, S ≥ 0.60, D ≥ 0.10, F ≥ 0.50, and M ≤ 97.00, while excluding those for which CDS sequences could not be retrieved based on the GFF files. MAFFT v7.525 [44] was employed for single-copy-sequence multiple sequence alignments (-genafpair-maxiterate 1000), and trimAl v1. 5 [45] was employed for trimming of sequences (−gt 0.85 −cons 30). A phylogenetic tree was constructed by IQtree v2.3.6 [46] (−m JTT+F+R10 −B 1000), multiple trees were merged using Astral v5.7.8 [46] (−m JTT+F+R10 −B 1000), multiple phylogenetic trees were merged with Astral v5.7.8 [47], and *Sphagnum magellanicum* was kept as the outgroup for animal phylogenetic analysis.

### 2.2. Preparation of Genome and Annotation Files

The genome and GFF should be properly formatted. To preprocess genome and annotation files, the longest candidates of the species were extracted via the AGAT Toolkit [48]. After GFF files were processed, GFF3 files that met the aforementioned conditions were refined. The agat_convert_sp_gxf2gxf. pl script was used to correct errors and deficient annotation in the GFF files. The longest isoform was kept by agat_sp_keep_longest_isoform. pl. Candidate species CDS and protein sequences were batch-retrieved by GffRead v.12.7 [49]. Subsequently, CDS and proteins of the candidate species were combined to single files.

### 2.3. Identification of the IL-10 and IL-10R Families

A database was created using pfam_stcan-1.6 [50], and members of distant species gene families were located using hmmer-3.3.2 [51]. To identify gene sets that contain several structural domains, we took the intersection of the search results for each structural domain. To obtain more precise gene family information, we additionally employed the BLASTp [52] identification method (−value 1 × 10^−5^). The intersection of the Pfam and Blastp identification results was used to determine the final identified gene set.

### 2.4. Conserved Motif Identification

Conservative domain analysis is a valuable technique for determining protein function, directing medication design, analyzing gene family evolution, and identifying biomarkers. To get the candidate sequence’s conservative sequence information, the R-script was used to extract motif, domain, and gene structure information from the pfam_stcan-1.6 [50] search results and gene location information in the GFF file.

### 2.5. Gene Family Clustering and Classification

Data visualization was conducted using R version 4.4.1. The package dgfr v0.0.0.9 [53] was utilized to ascertain the optimal number of k-means clustering groups for protein sequences (min_clust = 6, max_clust = 11), and to compute the average and median similarity within each cluster. The esthetics of phylogenetic trees were improved using ggtree v3.12 [54] and ggtreeExtra v1.14 [55]. Visualization and enhancement of motif, domain, and gene structure were achieved through the use of gggenes v.5.1 [56]. For visualizing sequence alignment, MSA v1.36.1 [57], employing the MUSCLE algorithm [58], was utilized. Gene subfamilies are identified and clustered in-depth using k-means clustering, motif, domain, and gene structure data, and gene subfamilies with apparent structural characteristics were separated. After the distance matrix is embedded into a 2D space using classical multidimensional scaling (MDS, cmdscale), cluster labels are assigned by k-means; the resulting coordinates are obtained via eigendecomposition, similar to PCA. Therefore, the term “PCA plot” will be used hereinafter, but strictly speaking, it is an MDS visualization.

### 2.6. Analysis of the IL-10 and IL-10R Family Variants

Seqlogo analysis was conducted using Weblogo v3.7.12 [59] on sequences from different groups. Structural prediction of reorganized protein sequences was performed using ESMfold v1.0.3 (https://github.com/facebookresearch/esm, accessed on 1 October 2025) with the pre-trained model esm2_t48_15B_UR50D [60]. Conservative structure alignment of predicted structures among all redefined groups was carried out using US-align [61]. Visualization and refinement of the structure alignment results were done using UCSF ChimeraX v1.9 [62].

### 2.7. dN/dS Analysis

In this study, the Single-Likelihood Ancestor Counting (SLAC) model [63] in HyPhy was used to calculate the ratio of nonsynonymous substitutions (dN) to synonymous substitutions (dS). The operation was performed in batches via command line to generate JSON format results. The JSON files were uploaded to hyphy-vision (http://vision.hyphy.org/, accessed on 1 October 2025) for visualization. dN/dS > 1 indicates positive selection, dN/dS = 1 indicates neutral evolution, and dN/dS < 1 reflects purifying selection.

## 3. Results

### 3.1. Identification and Distribution of IL-10s and IL-10Rs in the Animal Class

Based on the previous research, we constructed an evolutionary tree spanning approximately 1290 million years (Mya), encompassing 491 animal individuals in a phylogenetic tree, including 480 animal species and an outgroup of *Sphagnum magellanicum*. The species were ultimately selected ranging from higher to lower taxa, including Mammalia, Aves, Actinopteri, Lepidosauria, Reptilia, and other classes. After obtaining the intersection of gene sets identified by Pfam and BLASTp identification methods, we acquired an evolutionary tree of 2079 IL-10-like protein sequences from 408 individuals (403 species) (Figure 2A–C), and an evolutionary tree of 1674 IL-10R-like protein sequences from 412 individuals (406 species) (Figure 2E–G).

Based on the results of PCA clustering for all IL-10s, we combined Identity, unrooted trees, and the relationships between each IL-10 to manually group the IL-10 signaling system, dividing the IL-10s into five groups, namely Groups I to V (Figure 2C). Among them, the genes of the type III interferon IL-28 (IFNL1, 2, 3) subfamily are widely distributed in mammals, and the IL-28 subfamily genes in mammals have higher Identity compared to human IL-28s (Figure 2A). All genes of this subfamily are distributed in cluster 3 (Figure 2D) and are only distributed in mammals. It is the most conserved IL-10 subfamily with a later evolutionary history among IL-10s (Figure 2A–C). The most distant species in which homologous genes of this family are distributed can be traced back to Vombatus ursinus and Phascolarctos cinereus of Diprotodontia, and its Identity with human IL-28s (at the 19q13.2 locus) is between 50.81% and 54.45%. Of particular note is Cluster 4, which comprises nearly all IL-19, IL-20, and IL-24 homologs, along with a small subset of IL-10 homologs. Most of them have homology to each other (Figure 2A). Moreover, IL-10, IL-19, IL-20, and IL-24 are all located on the 1q32 locus, suggesting that these four genes may have originated from the same gene. Among the species with IL-10 homologous gene distributions, the farthest can be traced back to *Callorhinchus milii* (36.62%) during the Chondrichthyes period; among the species with IL-19, IL-20, and IL-24 homologous gene distributions, the farthest can be traced back to *Lepisosteus oculatus* (31.03–43.10%) during the Actinopteri period. IL-22 and IL-26 are located on the 12q15 locus and do not have homology to each other. Among the species with IL-26 homologous Gene distribution, the farthest can be traced back to *Pelodiscus sinensis* (48.77%) under Reptilia.

Among the species with an IL-22 homologous gene distribution, the farthest can be traced back to *Paramormyrops kingsleyae* (33.33%) under Actinopteri (Appendix A). We also adopted the same clustering and classification strategy for IL-10Rs and grouped the IL-10Rs into five groups, namely Groups I to V (Figure 2F). The results showed that among the species with IL-10RA/B and IL-20RA/B homologous gene distributions, the farthest can be traced back to *C. milii* (23.11–47.80%) under Cartilaginous fish (Figure 2H), while IL-22RA1/2 can be traced back to *P. kingsleyae* (40.69%) and (27.13%) under Actinopteri, respectively. Among them, IL-20RA and IL-22RA2 are located on the same site, 6q23.3, and blastp results implied that there is similarity between them (Figure 2E).

### 3.2. Conserved Motif, Domain, and 3D Structure of IL-10s and IL-10Rs Across Several Groups

The basic composition of IL-10 family receptors and ligands is shown in Figure 3: IL-10 groups I through III primarily consist of a single IL10 domain, with a small number in group III also containing IL22 domains; group IV consists entirely of a single IL22 domain; and group V proteins are composed of a single IL28A domain. IL-10R proteins in groups I to V generally consist of a Tissue_fac domain followed by an Interfer−bind domain; approximately half of the proteins in group III and a small number in group IV contain only a single Tissue_fac domain; a few members in group V contain more than one Tissue_fac/Interfer−bind domain. These two domains typically occur in repeats or at intervals, and these repeated domains may confer unique functions to the proteins in this group that distinguish them from other groups (Appendix A). IL-10 signaling system members exhibit high structural conservation. This conservation is attributed to the cross-species conserved IL10, IL22, and IL28A domains, as they are present in nearly 100% of IL-10s (Figure 3). Structural alignment analyses show that these regions remain largely unchanged. Although no obvious shared sequence features were observed in the SeqLogo analysis (Appendix A), it can be determined that despite significant variations in conserved sequences among different groups, their protein structures are extremely similar (6–7 α-helices).

Within IL-10, groups I and II consist of a single IL-10 domain, and group IV consists of a single IL-22 domain (Figure 3). All species in these three groups belong to Chordata, covering mammals, birds, reptiles, amphibians, and fish, and possess vertebrae and a central nervous system. Group V consists of a single IL-28A domain (Figure 3), and these species all belong to Mammalia, possessing the basic trait of mammals. These species represent multiple different orders and families within the class Mammalia, demonstrating the diversity of mammals, and they have diverged from a common ancestor, adapting to various ecological environments. Group III mainly contains IL-22 domains and IL-10 domains (Figure 3). All species containing IL-10 domains are members of the subphylum vertebrates within the phylum Chordata, covering numerous species from fish, amphibians, reptiles, and birds to mammals. Meanwhile, in the third group, only 12 species retained the IL-22 domain alone, and these species all belong to Mammalia, covering different evolutionary branches, such as Carnivora (e.g., meerkat), Proboscidea (e.g., African elephant), and Rodentia (e.g., ground squirrel) (Appendix A).

In IL-10Rs, all groups are composed of Tissue_fac and Interfer-bind domains, with roughly the same proportion of Tissue_fac and Interfer-bind domains in groups I, II, and V (Figure 3). All species in group I that contain both Tissue_fac and Interfer-bind domains belong to Amniotes, covering mammals, birds, reptiles, etc. (Appendix A). In group II, except for *Ophiophagus hannah* and *Echinops telfairi*, which only contain the Tissue_fac domain, all species containing both Tissue_fac and Interfer-bind domains belong to vertebrata, covering mammals, birds, reptiles, amphibians, and fish, etc. (Appendix A). All species in group V that contain both Tissue_fac and Interfer-bind domains belong to vertebrata, covering mammals, birds, reptiles, amphibians, and fish, etc. Among them, the number of bird and mammal species accounts for the vast majority in this group (Appendix A). All species in group III that contain both Tissue_fac and Interfer-bind domains are mainly birds, mixed with a small number of other vertebrates, while all species that contain only the Tissue_fac domain belong to Mammalia, covering all major orders (Appendix A). All species in group VI that contain both Tissue_fac and Interfer-bind domains are mainly birds and mammals, while species that contain only the Tissue_fac domain are mainly fish.

IL-10 signaling system members are highly conserved in structure. Studies have shown that OpIL-10 of *Oplegnathus punctatus* encodes 187 amino acids and possesses typical IL-10 signaling system characteristic motifs and a predicted α-helix structure [32], suggesting that the IL-10 gene family may have a distant evolutionary origin. OpIL-10 shares high similarity with IL-10 of other fish species, especially with IL-10 of *Notolabrus celidotus* and *Epinephelus Lanceolatus* [32]. This cross-species similarity further supports the conservation of the IL-10 gene family during evolution and suggests that it may have a common ancestral gene. The function of IL-10 in different species also appears to be highly conserved. For example, in *O. punctatus*, the expression of OpIL-10 is significantly upregulated in multiple immunization-related tissues after infection with *Vibrio harveyi* and SKIV [32]. Similarly, in *Nibea albiflora*, IL-10 was also significantly upregulated after Vibrio parahaemolyticus and *Vibrio alginolyticus,* and Poly I: C [33]. This functional similarity may reflect the importance and conservation of the IL-10 gene family in evolution. The study first demonstrated the existence of the IL-10 ligand-receptor system in *N. albiflora* [33]. This finding not only reveals the important role of IL-10 in the fish Immune system but also suggests that IL-10 and its receptor system may have formed early in evolution.

In IL-22, N54 was identified as an important binding site, which is subsequently transcriptionally modified to N-linked glycosylation [64]. Studies have identified several IL-10RA missense mutations associated with childhood or early onset IBD, including p.W45G, p.Y57C, p.W69G, p.T84I, p.Y91C, p.R101W, p.R117C, and p.R117H [65]. These mutations may affect the structural stability of IL-10RA and its binding affinity to IL-10, thereby interfering with the IL-10 signaling pathway.

IL-10R1 retains important amino acid residues related to signal transduction in the cytoplasmic domain, including two redundant peptide motifs GYXXQ. These motifs are essential for STAT3 recruitment and activation [66]. The binding IL-22R1 to IL-19 interface is formed by the long loop of IL-20R1 and the raised region of IL-19, exhibiting complementary charge properties. This binding interface is rich in aromatic residues [67]. When comparing the sequences of the three cytokines (IL-19, IL-20, IL-24) that bind to IL-20R1, significant conservation in the length of their interaction loops was found. A residue considered to play a key role in receptor binding specificity is a conserved Glutamic acid [67]. The genomic organization, protein structure, and receptor function of IL-10R1 are remarkably evolutionarily conserved in higher vertebrates [66]. For example, the duck *IL-10R1* gene has a structure of seven exons and six introns, similar to the chicken and human *IL-10R1* genes in exon size, but the avian gene is more compact [66]. One study found that when comparing human and mouse cell-derived Interleukin-10 (cIL-10) with the Epstein–Barr Virus *BCRF-I* gene product (viral Interleukin-10, vIL-10), Amino acid isoleucine at position 87 of cIL-10 was found to be critical for its immunostimulatory function. This finding suggests that vIL-10 may be a captured and selectively mutated *cIL-10* gene that contributes to viral pathogenicity, leading to ineffective host immune response [68]. The IL-10 receptor subfamily genes may have expanded through gene duplication and functional divergence. For example, one study reported the identification of a novel gene, *IFNLR1*, which encodes a 520-amino-acid transmembrane protein with 22% Amino acid homology to the extracellular domain of the IL-20 receptor. The *IFNLR1* gene is located on Chromosome 1, 25 kb away from the IL-22 receptor gene [69]. This gene arrangement suggests that the IL-10 receptor subfamily genes may have arisen through local gene duplication and functional divergence.

### 3.3. The Evolution Pattern of IL-10s and IL-10Rs

To explore the evolutionary patterns of genes encoding IL-10s and IL-10Rs, we plotted the distribution of copy numbers on the species tree, and the results showed that from ancient animals to mammals with a developed immune system today, both IL-10s and IL-10Rs have undergone minor copy number changes (Figure 4A,E). This is particularly prominent in mammals, where the gene numbers of IL-10s and IL-10Rs mostly exceed 5, and a few species reach 7, making them the most abundant class of IL-10s and IL-10Rs (Figure 4A,E). Unlike mammals, Aves have slightly more IL-10Rs than IL-10s, and a few Aves species can have as few as two IL-10s and IL-10Rs genes, a situation that is very rare in mammals. The number of IL-10Rs in Reptilia and Actinopteri is also slightly more than that of IL-10s. Most interestingly, in Actinopteri, which mainly consist of IV and V, the numbers of IL-10s and IL-10Rs show the opposite trend, with the number of ligands being much larger than the number of receptors. The number of genes encoding receptors and ligands in certain species, as well as the distribution of IL-10s and IL-10Rs in species, reflects their ancient origins. Even in ancient Bony fish, as shown in Figure 4B,F, some species have more than four receptor genes, which further supports the idea that the IL-10 Immune system is an ancient Immune system. However, receptor diversification did not occur until the emergence of terrestrial organisms, peaking in mammals.

## 4. Discussion

### 4.1. The Ancient Origin of IL-10s and IL-10-Rs

The IL-10 signaling system is an ancient family that has continuously differentiated and undergone adaptive changes during evolution. During evolution, IL-10 signaling system cytokines are widely presented in vertebrates, found in everything from fish to mammals. This study found that the core member IL-10 homologous gene can be traced back to Cartilaginous fish (with 36.62% similarity to humans); the IL-19/20/24 and IL-22/26 subfamilies can be traced back to Bony fish and reptiles, respectively; although the IL-28 subfamily is most conserved in mammals (with 50.81–54.45% similarity to humans), it is also distributed in Bony fish, and its overall origin dates back to the early evolution of vertebrates. IL-10RA/B and IL-20RA/B homologous genes can be traced back to Cartilaginous fish (with 23.11–47.80% similarity to humans); IL-22RA1/2 can be traced back to Bony fish, with its core framework formed approximately 450 million years ago, and some members (such as IL-20RA and IL-22RA2) originated from gene replication and differentiation. Homologous genes of IL-10s and IL-10Rs co-exist in ancient taxa (Chondrichthyes, Bony fish) and are subsequently co-retained in higher vertebrates; when IL-10s expand in mammals, IL-10Rs adapt through heterodimeric combinations, with their quantitative changes mutually adapting. They all bind as “R1+R2 type receptor subunit” heterodimers (e.g., IL-10 binds IL-10R1/IL-10R2), with IL-10R2 being a multi-ligand shared subunit; after binding, they uniformly activate the Jak/STAT pathway, and the core mechanism is conserved. The Tissue_fac+Interfer-bind domains of IL-10Rs precisely match the α-helix structure of IL-10s, and the binding interface residues are conserved; when IL-10s undergo functional differentiation, IL-10Rs structures are finely tuned, but the core framework remains unchanged.

IL-10 signaling system cytokines are a class of immunoregulatory molecules with diverse biological functions, exhibiting a certain degree of conservation and diversity during evolution. IL-10s and IL-10Rs originated from early vertebrates’ evolution, developing in a “common ancestor + gradual differentiation” model: IL-10s, through gene clustering, formed 3 gene clusters, with domains specializing according to functional grouping (e.g., Group V contains only the IL-28A domain); IL-10Rs expanded via gene duplication, with domain combinations adapting to ligands. Both synergize throughout the process, from ancient fish to mammals, maintaining high conservation in structure, binding patterns, and signaling pathways, while also adapting to different ecological and immune needs through species-specific fine-tuning (e.g., IL-22 functional differentiation), together forming the core immune regulatory system of vertebrates.

This study found that while the IL-28 subfamily exhibited the highest modern distribution and highest sequence conservation in mammals, subfamilies such as IL-10, IL-19/20/24, and IL-22/26 are ancient, originated in fish, and exhibited certain sequence conservation in a wider range of vertebrates. They all retained the basic structural framework as class II α-helix cytokines but varied in sequence details and specific distribution due to their evolutionary history. Studies have shown that fish IL-10 and IL-20L molecules are evolutionarily similar to mammals’ IL-10 signaling system members and exhibit high conservation in amino acid sequences and structures [70]. These conserved features include conserved motifs and specific helical structures, such as 6 α helices and 4 conserved cysteines in IL-10, and 5 α helices and 6 conserved cysteines in IL-20 [70]. Furthermore, the structure of IL-10 signaling system genes is also evolutionarily conserved, typically consisting of five coding exons and four phase 0 introns, and these genes are tightly linked in the genomes of corresponding species [70]. This conservation of gene structure supports the hypothesis that IL-10 subfamily cytokines share a common origin with IL-10. In different species, the expression patterns and functions of IL-10 signaling system cytokines also differ. For example, in fish, IL-10 and IL-20L may play a role in the host’s late-stage reaction to bacterial infection, which differs from the expression pattern in mammals [70]. Furthermore, IL-10 signaling system members also play important roles in regulating inflammatory reactions and immune responses, suggesting that they may have participated in the host’s defense mechanisms against pathogens during evolution [71].

In summary, the IL-10 signaling system of cytokines has maintained the conservation of structure and function during evolution while also exhibiting certain diversity in different species. These traits enable them to play an important role in immunization regulation and host defense.

### 4.2. IL-10s and IL-10-Rs Structure and Cross-Species Characteristics

This study used the HyPhy method to analyze the selection pressure of IL-10s and IL-10Rs, and then visually displayed the relationship between codon sites and dN-dS values by plotting a Site Graph scatter plot (Appendix A), with the abscissa representing the codon site (Site) and the ordinate representing the dN-dS value, which is used to quickly identify amino acid sites under selection pressure. This article provides a cross-species selection pressure dataset of IL-10s and IL-10Rs (Appendix A). By analyzing the selection pressure of IL-10s and IL-10Rs using the HyPhy method, the results showed that the two were affected by different selection mechanisms during evolution: IL-10s generally showed neutral or weak positive selection. Although its function is highly conserved, it is subject to positive selection pressure at certain sites, which may be involved in adaptability evolution to adapt to the needs of different species or immunization environments, while IL-10Rs showed the characteristics of balancing selection, with strong positive selection coexisting with strong negative selection at some sites. This balancing selection ensures that IL-10Rs can adapt to different pathogen, metabolic, environmental, therapeutic and other multi-dimensional pressures while maintaining signal transduction stability, reflecting the complexity of the receptor in the evolutionary process.

Domain analysis of IL-10s based on groups (Group I–V) showed that its domains have clear division of labor and are highly conserved, and there are significant specificities in the domain composition of different groups; Group I to III: The core structure is a single IL-10 domain (PF00726.22). Notably, Group III contains not only the canonical IL-10 domain but also a subset of proteins that include an IL-22 domain (PF14565.11), making it the only group that simultaneously incorporates both domain types. Group IV: The structure is highly specialized, and all proteins are composed of a single IL-22 domain, without other domain redundancy, which is highly matched with the functions of this group related to intestinal immunization and tissue repair. Group V: The exclusive domain is the IL-28A domain (PF15177.11), and all members contain this single domain, which is consistent with its evolutionary positioning as a type III interferon (antiviral core function).

All IL-10s belong to the class II α-helix cytokine family. IL-10 is a six-helix bundle dimer, while other members are mostly seven-helix bundle monomers. Key sites, such as the N54 glycosylation site of IL-22, are conserved across species to ensuring functions. The five groups of IL-10s have clear boundaries in species taxonomy, reflecting species-adaptive selection during evolution. Group I to IV cover chordates (mammals, birds, fish, etc.), including the complete lineage from lower vertebrates to higher vertebrates, indicating that the functions of these groups of IL-10s (such as basic immune regulation and inflammation control) are core immune mechanisms shared by vertebrates. Group V exists only in mammals, including multiple evolutionary branches, such as Carnivora, Proboscidea, and Rodentia, and the IL-28A domain protein of this group is strongly correlated with antiviral function, suggesting that mammals have formed a unique IL-28 subfamily regulatory mechanism in the immune evolution to cope with viral infection.

Despite the conserved structure of IL-10s, functional differentiation exists across different species, with the following typical examples. For instance, IL-22 exhibits clear functional divergence; in Muridae, it mainly exerts anti-bacterial effects by activating the innate defense mechanisms of intestinal epithelial cells and inhibiting the colonization of intestinal pathogens such as Escherichia coli. In humans, however, the core function of IL-22 shifts to immune regulation, participating in the maintenance of intestinal immune homeostasis, and is closely related to the pathogenesis of inflammatory bowel disease, such as Crohn disease. This difference may stem from the evolutionary differences in intestinal microecology and immune cell distribution between humans and Muridae. For functional conservation and adaptation of IL-10, the anti-inflammatory function of IL-10 is highly conserved across species. For example, after Bony fish (*O. punctatus*) are infected with *Vibrio harveyi*, the expression of its IL-10 (OpIL-10) in immune-related tissues, such as the spleen and kidney, is significantly upregulated, protecting tissues by inhibiting excessive inflammation, while in mammals, IL-10 has evolved a new function of promoting the survival of regulatory T cells (Treg) in addition to anti-inflammation, which is an important manifestation of the complexification of the adaptive immune system in mammals.

The domains of IL-10Rs are characterized by a “basic combination + group specialization”. All groups have the Tissue_fac domain (PF01108.22) and Interfer-bind domain (PF09294.15) as their core, but there are differences in the number or integrity of domains in different groups: Group I, II, V; the domain composition is the most complete and balanced, and the proportion of Tissue_fac and Interfer-bind domains in the protein is approximately equal (approximately 48–55%), with no single domain deletion. This complete structure is the basis for ensuring efficient receptor-ligand binding and complete signal transduction (e.g., the IL-10R1/IL-10R2 complex requires two types of domains to synergistically recognize the IL-10 ligand). For Group III and IV, there is domain simplification, in which 50% of Group III proteins and a small number of Group IV proteins contain only a single Tissue_fac domain, with Interfer-bind domain deletion. These receptors may indirectly achieve functional complementation by forming heterodimers with other receptor subunits containing complete domains. For the special structures of Group V, a few members contain multiple Tissue_fac and/or Interfer-bind domains (e.g., 2 Tissue_fac+1 Interfer-bind). The repeated domains may enhance the binding specificity or affinity of the receptor to the ligand, enabling the receptors in this group to exhibit higher reaction efficiency in the antiviral signaling pathway (e.g., IL-28A/B binding to IL-28R/IL-10R2).

The intracellular signaling-related structures of receptors are highly conserved across species, which is the core to ensure the activation of the Jak/STAT pathway. A key intracellular motif within IL-10R1, the GYXXQ peptide sequence, is retained in duplicate within the cytoplasmic domain across species, such as humans, ducks, and mouse, which is a key binding site for STAT3 protein recruitment and phosphorylation. If this motif is mutated, such as an amino acid substitution, it will directly lead to the interruption of the IL-10-mediated immunosuppression signal. The conservation of this motif has not changed from Cartilaginous fish, such as *C. milii.*, to mammals. Taking the binding of IL-20R1 to IL-19 as an example, the binding interface between the two is formed by the long loop region of IL-20R1 and the raised region of IL-19, and the interface is rich with aromatic residues, such as Phenylalanine and Tyrosine, and there are also conserved Glutamic acid residues. This residue is highly consistent in the sequences of IL-19, IL-20, and IL-24 (all of which bind to IL-20R1), and is a “molecular switch” that determines the specificity of receptor-ligand binding.

The grouping (Group I-V) of IL-10Rs is highly associated with the evolutionary status of species, such as Amniota and vertebrates, reflecting the synergistic relationship between receptor evolution and species adaptability. In Group I, species are strictly limited to Amniotes, covering mammals (humans, pigs), birds (chickens, ducks), reptiles (crocodiles, turtles), with no amphibians or fish members. Amniotes are a relatively advanced group of vertebrates. This group of receptors may be a specific receptor group evolved to cope with complex terrestrial pathogens, such as bacteria and fungi, during the terrestrial adaptation of amniotes. In Groups II and V, species cover the widest range, all of which are vertebrata, including mammals, birds, reptiles, amphibians (frogs), and fish (Bony fish and Chondrichthyes). Among them, avian and mammalian species account for over 80% in Group V, which are the functionally dominant groups in this group. In Group II, except for *O. hannah* and *E. telfairi*, which contain only the Tissue_fac domain, the remaining species contain both complete types of domains, reflecting the universality of basic receptor function in vertebrates. In Group III and IV, the species composition presents a “group-dominated” trait. In Group III, species containing both types of domains are mainly birds (accounting for approximately 70%), mixed with a small number of reptiles and mammals, while species containing only the Tissue_fac domain are all mammals, and cover all major orders (carnivora and pimates). In Group IV, species containing complete domains are mainly birds and mammals, while species containing only the Tissue_fac domain are mainly fish (*P. kingsleyae* in Bony fish), and this differentiation may be related to the immunization needs of different groups, such as pathogen defense in the aquatic environment for fish.

### 4.3. The Association of IL-10s and IL-10Rs with Diseases

IL-10′s immunomodulation is a phylogenetically ancient mechanism that induces tolerance and fine-tunes the inflammatory reaction; it is observed not only in mammals but in all vertebrates, including birds, amphibians, and fish [72]. IL-10 is an anti-inflammatory cytokine that functions by binding to the IL-10 receptor (IL-10R), modulating the immune response and suppressing inflammation, an interaction that has important implications for the pathogenesis and treatment of multimorbidity.

IL-10s and IL-10Rs play important roles in the occurrence and development of multimorbidity. IL-10 is an immunomodulatory cytokine that inhibits inflammatory reactions. The epigenetic regulation of *IL-10* gene expression involves multiple complex mechanisms, including chromatin remodeling, 3D chromatin loop formation, histone modifications (such as histone acetylation and methylation), and DNA methylation and other epigenetic modifications; these mechanisms collectively and precisely regulate the expression level of IL-10 [73]. In human macrophages, analysis by ATAC-seq and ChIP-seq revealed that LPS stimulation can significantly increase the chromatin accessibility and histone acetylation level of macrophage-specific enhancers in the *IL-10* gene locus, among which the p300 protein is recruited to these cis-regulatory elements, and the dynamic binding of transcription factors PU.1, AP-1, and STAT3 regulates the expression of IL-10 [74]. In addition, epigenetic mechanisms also participate in the regulation of IL-10 through microRNAs (such as miRNA) and long non-coding RNAs (such as GAS5 lncRNA), which can regulate the expression of IL-10 by targeting mRNA or affecting chromatin structure [75]. The downstream epigenetic regulation of IL-10 mainly activates transcription factors through its signaling pathway, and these transcription factors then recruit epigenetic modification enzymes to specific gene loci, and regulate the expression of downstream genes by changing chromatin status and histone modification patterns, thereby achieving fine regulation of immune response. At the same time, changes in histone modifications, especially histone acetylation, may also lead to abnormal expression of IL-10 mRNA. In addition, IL-10-related microRNAs (miRNAs) are abnormally expressed in multimorbidity, and the long-chain non-coding RNA (lncRNA) growth arrest-specific transcript 5 (Gas) can also inhibit the expression of IL-10. In different diseases, DNA methylation modification of the *IL-10* gene locus can affect the occurrence and development of diseases. In addition, miRNA-98 is an important miRNA that is associated with IL-10 in multimorbidity, such as HCC, lung cancer, and myocarditis [75]. Although IL-10 has well-known anti-inflammatory effects, IL-10-producing T cells have dual functions, both promoting and controlling inflammatory reactions, and this seemingly contradictory function may be related to cell source and microenvironment. The heterogeneity of IL-10-producing T cells and their effects on target cells have been a research hot topic in recent years. Studies have shown that IL-10 can support the activation of B cells and CD8+ T cells, suggesting the versatility of IL-10 in immunization responses [76].

Dysregulation of IL-10 and IL-10R pathways is an essential factor in the pathogenesis of multimorbidity, and downregulation of IL-10/IL-10R signaling is mainly associated with the occurrence of inflammatory and allergic diseases [75], including rheumatoid arthritis (RA) [77], acute respiratory distress syndrome (ARDS) [78], coronary artery disease (CAD) [79], Henoch-Schonlein purpura (HSP) [80], etc. For example, IL-10 plays a key role in the protective mechanism of allergic diseases (especially asthma disease); studies have shown that IL-10 is located downstream of IL-6 and is a key mediator required for Acinetobacter lwoffii (AL)-induced asthma disease protection, rather than IL-17; AL triggers local pro-inflammatory responses in the respiratory tract, leading to a sustained increase in systemic IL-6 levels, thereby promoting the production of IL-10 in CD4+ T cells; in *IL-10* gene knockout mice, AL can no longer provide asthma disease protection, confirming the decisive role of IL-10 in this protective mechanism [81], while upregulation of IL-10 signaling may be associated with autoimmune diseases and cancer [75], including Systemic Lupus Erythematosus (SLE) [82], Hepatocellular carcinoma of liver (HCC) [83], etc. This duality makes the IL-10 and IL-10R pathways a complex but important target in immunotherapy, requiring precise regulatory strategies. Targeting strategies against IL-10s and IL-10Rs are constantly evolving, providing new hope for the treatment of multimorbidity.

Researchers are developing various targeted strategies and drugs for inflammatory diseases and cancer caused by dysregulation of IL-10 and IL-10R pathways. These strategies aim to restore the balance of the pathway, rather than completely blocking or activating it. As studies have found, the biologically active form of IL-10 is an unstable homodimer with a short half-life and is easily degraded in vivo. To improve the therapeutic efficacy of IL-10, researchers have developed a novel IL-10 dimer by connecting two IL-10 monomers via a linker to form a stable hetero- or homodimer, which can significantly enhance its stability and biological activity and prolong its action time in vivo. This novel IL-10 dimer exhibits better temperature and pH-dependent biological stability and more effectively inhibits inflammatory reactions than natural IL-10 [84]. Studies have also found that cellular interleukin 10s (cIL-10s) of human and murine origin (cIL-10) are highly homologous in sequence and structure with the Epstein–Barr virus *BCRF-I* gene product (vIL-10). However, vIL-10 lacks the immunostimulatory effect of cIL-10 on certain cell types. The isoleucine at position 87 of cIL-10 is crucial for its immunostimulatory function; when isoleucine at this position is replaced by alanine, the immunostimulatory activity of cIL-10 is significantly reduced or even lost. This replacement is identical to the amino acid at that position in viral interleukin 10 (vIL-10). This finding reveals that the functional differences between cIL-10 and vIL-10 may stem from a single amino acid change, thus providing important clues for understanding the biological function of IL-10 and its role in immune regulation [68]. Based on computer analysis of the complex structure of IL-10 and the 1L-10 receptor ligand-binding subunit IL-10Ra, some research optimized a peptide composed of 20 amino acids, named NK20a, whose binding ability was confirmed by in vitro biophysical bilayer interferometry and cell experiments. The IL-10 inhibitory peptide exerted an anticancer effect on lymphoma B cells and was able to eliminate the inhibitory effect of 1L-10 on macrophages; the computer-designed NK20a has improved IL-10 receptor binding affinity and can serve as a tool for developing anticancer strategies [85].

The relationship between IL-10s and IL-10Rs with diseases transforms the evolutionary conservation (domains, pathways, key motifs) and species-specific functional divergence of the IL-10 signaling system into a target basis for clinical intervention: it not only verifies the explanatory value of evolutionary studies for disease mechanisms (such as inflammatory bowel disease, autoimmune diseases) through the conservation of key amino acid sites (e.g., mutations in the GYXXQ motif of IL-10R1 leading to STAT3 signaling defects, and the N54 glycosylation site of IL-22 affecting barrier repair), but also reveals the evolutionary roots of differences in disease spectra based on species-specific functional divergence (e.g., IL-22 is anti-bacterial in mice but associated with immunization regulation in humans); meanwhile, it transforms molecular mechanisms (bidirectional pathological effects of pathway dysregulation) into a target basis for drug development (such as peptide inhibitor, long-acting IL-10 dimer, anti-IL-22 monoclonal antibody), and builds a translational bridge from basic evolutionary research to clinical precision therapy (such as antiviral-targeted drugs) and animal breeding (selection of disease-resistant traits).

## 5. Conclusions

This study, through cross-species comparative genomics and phylogenetic analysis, for the first time revealed the synergistic evolutionary framework of the IL-10 signaling system (including ligands IL-10s and receptors IL-10Rs) in vertebrates. IL-10s originated from early vertebrates (from cartilaginous fish to bony fish), with their core domains (e.g., IL-10, IL-22, IL-28) being highly conserved, but achieving diverse immune regulation through gene replication and functional differentiation: the IL-10 subfamily dominates anti-inflammatory responses, the IL-20/22 subfamily mediates tissue repair, and the IL-28 subfamily specializes in antiviral functions in mammals. IL-10Rs, as type II cytokine receptors, can be traced back to cartilaginous fish (approximately 450 Mya), and their conserved Tissue_fac/Interfer-bind domain precisely interacts with the type II alpha helix structure of IL-10s, responding to ligand diversity through heterodimeric combinations (e.g., IL-10R1/R2, IL-22R1/IL-10R2); in mammals, receptors expand through domain duplication (e.g., Group V containing multiple Tissue_fac domains) and gene replication, achieving functional adaptation with subfamilies such as IL-28. Key motifs (e.g., GYXXQ of IL-10R1) mediate the STAT3 signaling pathway, whose mutations lead to immunodeficiency (e.g., IBD), confirming that coevolution of the ligand–receptor interaction interface is a core mechanism driving immune homeostasis. This evolutionary framework not only elucidates the adaptive refinement process of the IL-10 system throughout history but also provides a theoretical basis for targeted therapies (e.g., anti-IL-22 monoclonal antibodies) and animal disease-resistant breeding.

## Figures and Tables

**Figure 1 genes-16-01243-f001:**
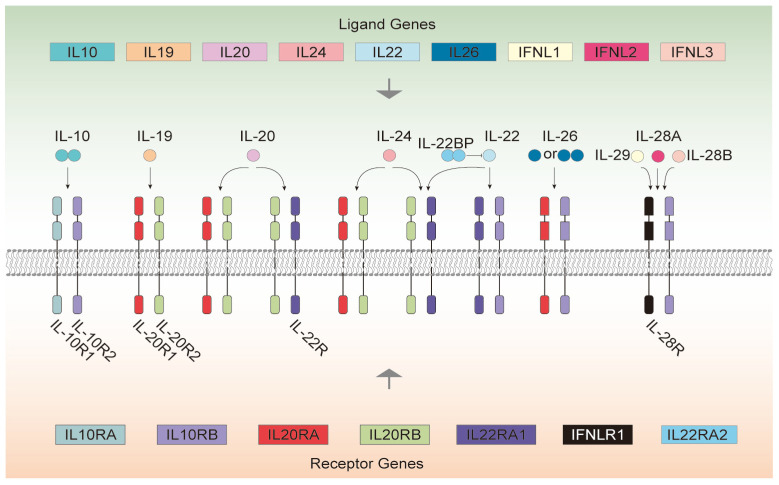
The human IL-10 signaling system comprises ligands, antagonists, and receptors, with color coding indicating genes or subunits derived from the same evolutionary origin. The product of the *IL22RA2* gene, IL-22BP, inhibits the binding of IL-22 to its receptor complex. By competitively binding to IL-22, it restricts the cytokine’s interaction with the functional receptor complex IL-22R/IL-10R2.

**Figure 2 genes-16-01243-f002:**
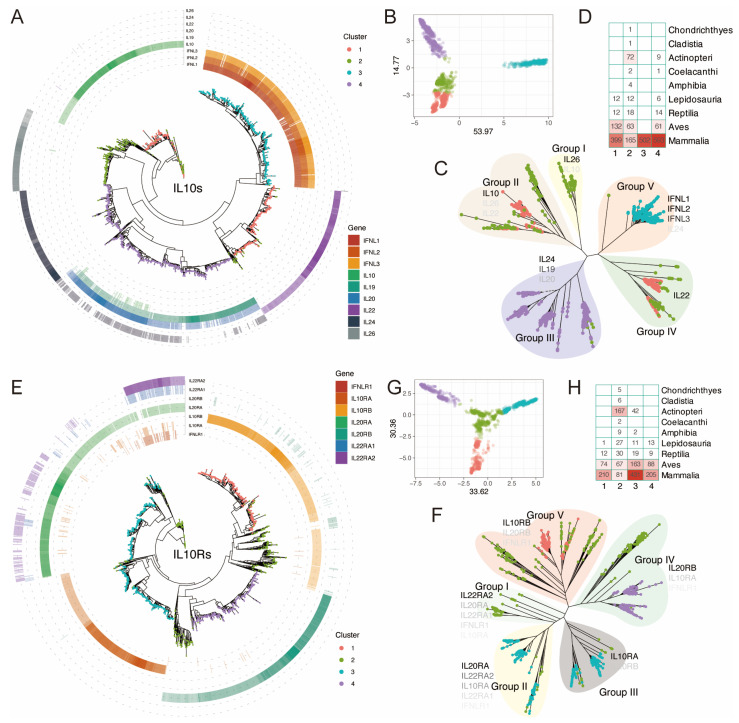
Phylogenetic investigation of the IL-10s and IL-10Rs across over 400 animal species. (**A**,**E**): The phylogenetic tree of the IL-10s and IL-10Rs is over 400 animal species, with different colored tip points representing different clusters and different colored rings outside the tree representing homologous genes (Appendix A), and the depth of color corresponding to the Identity (%) with homologous genes. (**B**,**G**): A PCA presents the total number of detected clusters in the IL-10s and IL-10Rs. Based on the optimal cluster numbers determined by k-means clustering and PCA of IL-10s and IL-10Rs, each cluster showed significant differences in the original trait space. (**C**,**F**): Phylogenetic tree and dispersion pattern of the IL-10s and IL-10Rs. The main genes of each group are marked in color fonts of different shades; the darker the color, the higher the proportion. (**D**,**H**): The distribution of distinct clusters results from k-means clustering of IL-10s and IL-10Rs. The size of the numbers and the color intensity represent the quantity of homologous proteins/genes of the same class, and the darker the color, the higher the proportion.

**Figure 3 genes-16-01243-f003:**
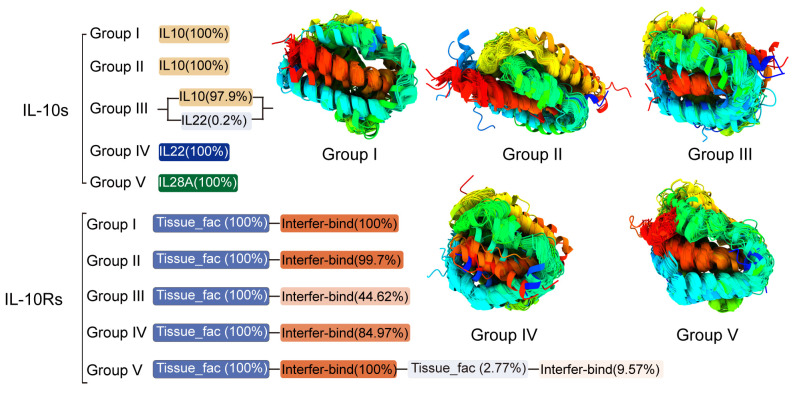
Domain architectures of IL-10s and IL-10Rs across over 400 animal species. Different colors represent distinct domain types, with color intensity indicating the relative abundance of each domain within the total sequences of its group; darker shades denote higher abundance. The protein structural alignment of IL-10s (non-conserved regions set to 100% transparency) displays a color gradient from red at the N-terminus to purple at the C-terminus. For further details, refer to Appendix A.

**Figure 4 genes-16-01243-f004:**
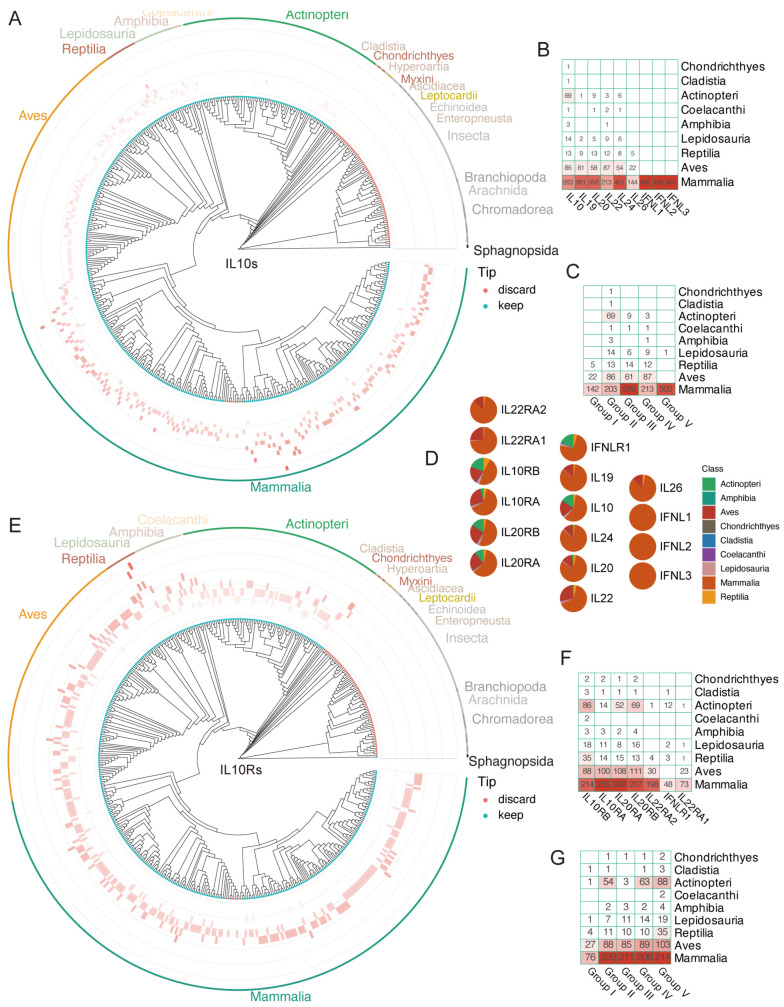
Evolutionary conservation and diversification of IL-10s and IL-10Rs across animal species. (**A**,**E**): Gene copy number dynamics of IL-10s and IL-10Rs in >400 animal species. IL-10s exhibited significant expansion during mammalian evolution, whereas IL-10Rs maintained stable copy numbers across all evolutionary stages. Cyan dots indicate species retained after BUSCO assessment; red dots denote species excluded due to data quality issues (absence does not imply gene loss). Colored arcs in the outermost layer represent taxonomic classes. (**C**,**G**): Homolog gene counts of IL-10s and IL-10Rs at the class level after phylogenetic clustering. The size of the numbers and the color intensity represent the quantity of homologous proteins/genes of the same class, and the darker the color, the higher the proportion. (**D**): Proportional distribution of IL-10s and IL-10Rs genes within each class. (**B**,**F**): Heatmap of PCA clustering distributions for IL-10s and IL-10Rs at the class level. The size of the numbers and the color intensity represent the quantity of homologous proteins/genes of the same class, and the darker the color, the higher the proportion.

**Table 1 genes-16-01243-t001:** Interleukin IL-10 signaling system members: expressing cells, receptors, signal activation, and biological functions.

Member	Expressing Cells	Receptors	Signal Activation	Biological Functions
IL-10	T lymphocytes, monocytes, and B cells	IL-10R1/IL-10R2	STAT1, STAT3, STAT5	Immunosuppressive, anti-inflammatory [15,19,20]
IL-19	monocytes	IL-20R1/IL-20R2	STAT3, STAT5	Immune regulation [15,19,21]
IL-20	Skin keratinocytes	IL-20R1/IL-20R2	STAT3, STAT5	Skin development and inflammation, hematopoiesis [15,19,22]
IL-22	Activated T cells and NKT cells	IL-22R/IL-20R2, IL-22R/IL-10R2	STAT1, STAT3, STAT5	Acute phase reaction, innate immunity [15,19,23]
IL-24	Melanocytes, Th2 cells, and monocytes	IL-20R1/IL-20R2, IL-22R/IL-20R2	STAT3, STAT5	Apoptosis, epidermal function, and the inflammatory cascade [15,19,24]
IL-26	memory Th1 cells, Th17 cells and monocytes	IL-20R1/IL-10R2	STAT1, STAT3	Mucosal and skin immunity [15,19,25]
IL-28; IL-29	monocytes	IL-28R/IL-10R2	STAT1, STAT2	Antiviral immunity [15,19,26]

## Data Availability

The original contributions presented in the study are included in the article/Appendix A. Further inquiries can be directed to the corresponding authors.

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
