# Peer review of "A Phylogenetic Perspective on the Evolutionary Patterns of the Animal Interleukin-10 Signaling System"

_genes, 2025, doi:10.3390/genes16111243_

Round 1

Reviewer 1 Report

Comments and Suggestions for Authors

Tang et al’s paper entitled “A Phylogenetic Perspective on the Evolutionary Patterns of the Animal Interleukin-10 Family” was informative and provided much needed analysis about the evolutionary relationships of the IL-10 family. The introduction provided an extensive overview of the IL-10 family and its signalling, whilst the discussion was thoroughly written. However the section “The Association of IL-10s and IL-10Rs with Diseases” felt out of place in a manuscript primarily about the molecular evolution of the IL-10 family.

Many of the figures provided information that was hard to interpret. Particularly Figure 4A and 4E which appears to have text, but it is illegible because the text is so tiny. Perhaps showing the data in a different manner would be more helpful for the reader, such as a summary figure or condensed phylogenetic tree.

It appears that some of the analysis described in the results were not shown. For example synteny analysis was described but not shown in a figure.

Crucially however, there were no sequence alignment to demonstrate similarities or differences between IL-10 family members. In particular, Section 3.2 would benefit from alignments of proteins from key species (for example one from mammals, aves, teleosts etc…). 

Moreover, non-synonymous to synonymous (dN/dS) analysis would be beneficial for this paper as it would be interesting to investigate where there may be selective pressure on each protein for each group of animals.

Finally, there were many more mammalain sequences listed, was this due to the avaialble dataset?

Overall, this was a well written paper, but more attention for how the reader can interpret the data needs to be addressed.

Author Response

[Oct 11, 2025] 

Dear Editors and Reviewers:

Thank you for giving us the opportunity to submit a revised draft of the manuscript “A Phylogenetic Perspective on the Evolutionary Patterns of the Animal Interleukin-10 Family (manuscript: genes-3898331)” for publication in genes. We appreciate the time and effort that you and the reviewers dedicated to providing feedback on our manuscript and grateful for insightful comments and valuable improvements to our paper. We have incorporated these comments made by reviews. Other changes are highlighted in the manuscript. Please see below, in blue, for a point-by-point response to the reviewers’ comments and concerns. All page numbers refer to the revised manuscript file with tracked changes. Furthermore, we would like to show the details as follows:

Comments 1: The introduction provided an extensive overview of the IL-10 family and its signalling, whilst the discussion was thoroughly written. However, the section “The Association of IL-10s and IL-10Rs with Diseases” felt out of place in a manuscript primarily about the molecular evolution of the IL-10 family.

Response 1: Thank you for your advice. We believe “The Association of IL-10s and IL-10Rs with Disease” section translates the evolutionary conservation of the IL-10 family (domains, pathways, key motifs) and species-specific functional differentiation into a rationale for clinical intervention targets. And relevant content was added to lines 726-739 of the manuscript. This section not only validates the guiding value of evolutionary theory for medicine but also provides direct translational pathways for targeted therapies and animal breeding, serving as a critical nexus bridging fundamental research to practical applications in the full text. Therefore, we consider this section to be of utmost importance to the entire paper. Please see the revised manuscript.

Comments 2: Many of the figures provided information that was hard to interpret. Particularly Figure 4A and 4E which appears to have text, but it is illegible because the text is so tiny. Perhaps showing the data in a different manner would be more helpful for the reader, such as a summary figure or condensed phylogenetic tree.

Response 2: Thank you for pointing out this. We carefully revised the text problems in Figures 4A and 4E. We pruned the species names that were not significant for interpreting the data meaning and zoomed in on the class names to enhance the sharpness and readability of Figure 4. The revised Figure 4 has been inserted on line 436 of the manuscript. Please see the revised manuscript.

Comments 3: It appears that some of the analysis described in the results were not shown. For example, synteny analysis was described but not shown in a figure.

Response 3: Thank you for your advice. We found that cross-species collinearity analysis of animals was not very helpful for this article, because compared with plants, existing tools are not effective in collinear in animals, and we have abandoned this part.

Comments 4: Crucially however, there were no sequence alignment to demonstrate similarities or differences between IL-10 family members. In particular, Section 3.2 would benefit from alignments of proteins from key species (for example one from mammals, aves, teleosts etc…).

Response 4: Thank you for your advice. To demonstrate the similarities or differences between IL-10 family members, we created SeqLogo diagrams, please see the Supplementary Figure 1.

Comments 5: Moreover, non-synonymous to synonymous (dN/dS) analysis would be beneficial for this paper as it would be interesting to investigate where there may be selective pressure on each protein for each group of animals.

Response 5: Thank you for your advice. We performed non-synonymous to synonymous mutation ratio (dN/dS) analysis. The methodology for this analysis has been incorporated into lines 246-252 of the manuscript, and the results of the dN/dS analysis among IL-10 family members are now presented in lines 527-542. Please see the revised manuscript.

Comments 6: Finally, there were many more mammalain sequences listed, was this due to the avaialble dataset?

Response 6: Thank you for pointing this out. This study mainly relies on public genome databases (NCBI, Ensembl, etc.), and among the currently available public genome data, there are the most mammals’ sequences, followed by birds and bony fish sequences, therefore this study lists more mammals’ sequences.

Reviewer 2 Report

Comments and Suggestions for Authors

With interest I read the manuscript entitled ”A Phylogenetic Perspective on the Evolutionary Patterns of the Animal Interleukin-10 Family” (Manuscript ID: genes-3898331), written by Tang and colleagues. Although in principle I am not a very big fan of purely bioinformatics works, this time I find this work quite valuable and necessary. I have minor comments only:

1.        Please, verify the names of genes at https://www.ncbi.nlm.nih.gov/gene/. Then must be written in italics.

2.        In continuation, there are ales between-species differences in nomenclature, so make it clear that you use names of human genes in the manuscript.

3.        Please, add references to Table 1.

4.        By occasion, also mast cells produce IL-10 (PMID: 34067047), which should be mentioned.

5.        Please, be careful in writing, e.g. ”L-29” in Table 1.

6.        Lines 545-604. The relationships with diseases must be supported with references, e.g. for allergic diseases PMID: 28322581, 36458896, etc.

7.        Lines 554-555. What you write here is a bit naïve, I mean the first general part of the sentence. Please, describe epigenetic regulation of the IL10 gene expression. Please, describe how epigenetic mechanisms are involved in the epigenetic regulation downstream of IL-10.

8.        By occasion, in the same sentence, why is it ”Protein” not ”protein”. There are many other words written this way throughout the manuscript. Please, correct.

9.        Line 1. What is the type of this manuscript?

10.  Graphical abstract should be considered.

Author Response

[Oct 11, 2025]

Dear Editors and Reviewers:

Thank you for giving us the opportunity to submit a revised draft of the manuscript “A Phylogenetic Perspective on the Evolutionary Patterns of the Animal Interleukin-10 Family (manuscript: genes-3898331)” for publication in genes. We appreciate the time and effort that you and the reviewers dedicated to providing feedback on our manuscript and grateful for insightful comments and valuable improvements to our paper. We have incorporated these comments made by reviews. Other changes are highlighted in the manuscript. Please see below, in blue, for a point-by-point response to the reviewers’ comments and concerns. All page numbers refer to the revised manuscript file with tracked changes. Furthermore, we would like to show the details as follows:

Comments 1: Please, verify the names of genes at https://www.ncbi.nlm.nih.gov/gene/. Then must be written in italics.

Response 1: Thank you for your advice. To enhance the standardization and professionalism of the overall paper, we have at https://www.ncbi.nlm.nih.gov/gene/ to verify and gene names written in italics. Please see the revised manuscript.

Comments 2: In continuation, there are ales between-species differences in nomenclature, so make it clear that you use names of human genes in the manuscript.

Response 2: Thank you for your advice. To enhance the standardization and professionalism of the overall paper, we clearly used the names of human genes in the manuscript and unified the naming throughout the entire text. Please see the revised manuscript.

Comments 3: Please, add references to Table 1.

Response 3: Thank you for your advice. To enhance the standardization and professionalism of the overall paper, we have added references in Table 1 on line 97 of the manuscript. Please see the revised manuscript.

Comments 4: By occasion, also mast cells produce IL-10 (PMID: 34067047), which should be mentioned.

Response 4: Thank you for your advice. We added this section, incorporating citations and arguments, to lines 107-115 of the manuscript. Please see the revised manuscript.

Comments 5: Please, be careful in writing, e.g. “L-29” in Table 1.

Response 5: Thank you for your advice. We carefully revised the format issue, changing “L-29” in Table 1 on line 97 of the manuscript to “IL-29” to enhance the clarity and readability of the article. Please see the revised manuscript.

Comments 6: Lines 545-604. The relationships with diseases must be supported with references, e.g. for allergic diseases PMID: 28322581, 36458896, etc.

Response 6: Thank you for your advice. In order to improve the standardization and professionalism of the full text, we added references corresponding to the relationship between IL-10 and various diseases in lines 684-691 of the manuscript. Also included are examples of how IL-10 plays a key role in the protective mechanisms of allergic diseases, especially asthma disease. Please see the revised manuscript.

Comments 7: Lines 554-555. What you write here is a bit naïve, I mean the first general part of the sentence. Please, describe epigenetic regulation of the IL10 gene expression. Please, describe how epigenetic mechanisms are involved in the epigenetic regulation downstream of IL-10.

Response 7: Thank you for your advice. We describe how the epigenetic regulation of IL10 gene expression and epigenetic mechanisms are involved in the downstream epigenetic regulation of IL-10 in pages 646-662 of the manuscript. Please see the revised manuscript.

Comments 8: By occasion, in the same sentence, why is it “Protein” not “protein”. There are many other words written this way throughout the manuscript. Please, correct.

Response 8: Thank you for your advice. We have carefully revised the format issue to improve clarity and readability in the article. Please see the revised manuscript.

Comments 9: Line 1. What is the type of this manuscript?

Response 9: Thank you for pointing out this. We have marked the type of this manuscript as "Article" on the first line of the manuscript. Please see the revised manuscript.

Comments 10: Graphical abstract should be considered.

Response 10: Thank you for your advice. Based on a careful Assessment of the core content of the paper, we believe that this article does not require an additional graphical abstract for the following reasons: This article has already completely presented the core evolutionary trajectory of the IL-10 family from its origin in vertebrates to functional differentiation through multi-dimensional visualization methods such as phylogenetic tree (Figure 2), domain conservation analysis (Figure 3), and sequence alignment (Figure 4). These figures have accurately covered key conclusions (such as subfamily differentiation, conserved motifs, species-specific adaptations), and the text Abstract has concisely summarized the research background, methods, results, and significance. As an independent supplementary module, the Information density of the graphical abstract significantly overlaps with existing figures, and it cannot provide additional value beyond the existing analysis.Therefore, to avoid visual redundancy and focus on core data presentation, we decided not to create a graphical abstract, ensuring that readers can efficiently grasp the full scope of the research through the main text figures and written summary alone.

Reviewer 3 Report

Comments and Suggestions for Authors

Review comments on genes-3898331

Manuscript ID: genes-3898331

Title: A Phylogenetic Perspective on the Evolutionary Patterns of the Animal Interleukin-10 Family

Authors: Liu Tang, Zeyu Zhou, Weibin Wang, Dawei Li, Tingting Hao*, Yue Chen*

Animal Genetics and Genomics

Major comments:

  • In this study, authors conducted large-scale comparative and phylogenetic analyses of IL-family and IL-10 receptor family from over 400 animal species. The results are somewhat interesting, but this manuscript cannot be acceptable for publication in “Genes” due to following reasons.
  • The IL-10 family is one of the important types of cytokines (including IL-10, IL-20, IL-22, IL-24, and IL-26, and possibly IL-28 and IL-29), that can stop the inflammation. Generally, these cytokines have a helical structure of homodimers. On the other hand, the IL-10 receptors (IL-10Rs)include cell surface receptor complexes that mediate the activity of the IL-10 family cytokines. These receptors are members of the Class II cytokine receptor family and consist of a specific ligand-binding α-chain and a shared β-chain (e.g., IL-10R1 and IL-10R2) that form heterodimers to transduce signals via the JAK-STAT pathway, initiating the receptor's anti-inflammatory role. Thus, IL-10 and IL-10R belong completely different protein families. Usually, a protein family is designated as a group of evolutionarily related proteins and, in most cases, a protein family has a corresponding gene family, in which each gene encodes a corresponding protein with a 1:1 relationship. Therefore, based on the terminology, it is very inappropriate to use the author-defined “interleukin-10 (IL-10) family”, which is comprising ligands (IL-10s) and receptors (IL-10Rs) (page 1, lines 13-14).
  • If authors were trying to investigate the interrelationship between the IL-10 family and the IL-10 receptor (IL-10R) family during the coevolution, the methods used in the present study were not inadequate. Since detailed investigations to clarify the co-evolution of the ligands and the receptors were not conducted in the present study, the parallel descriptions on each family are hard to follow.
  • The descriptions in “Conclusions” (page 15, lines 606-620) and in “Results and Conclusions” of Abstract (page 1, lines 19-27) are so different each other. In “Conclusions” (page 15, lines 606-620), only the results for IL-10 family are described. What are the authors’ conclusions?

Minor comments:

  • “(20)” (page 3, line 82) should be “[20]”?.
  • “PCA clustering” (page 7, line 146) is used without prior definition. PCA might be standing for Principal Component Analysis. It seems that more detailed descriptions on the analysis would be required in Materials and Methods section (page 6, lines 206-215).
  • In Figure 2B and G, no detailed description was shown in the legends for each PCA analysis (page 7, line 241-242).
  • “ALL” (page 7, line 251) should be “All”.
  • In Figure 2C and H (page 7) and in Figure3 (page 9), based on the phylogenetic tree analyses on IL-10 protein family and IL-10R protein family, authors showed groupings into 5 groups (Group I-V) for each family. I am not sure whether each group has some relationship each other if the group number was same?
  • “alpha” (page 12, lines 433,434) should be “α”

Comments on the Quality of English Language

The English could be improved to more clearly express the research results.

Author Response

[Oct 11, 2025]

Dear Editors and Reviewers:

Thank you for giving us the opportunity to submit a revised draft of the manuscript “A Phylogenetic Perspective on the Evolutionary Patterns of the Animal Interleukin-10 Family (manuscript: genes-3898331)” for publication in genes. We appreciate the time and effort that you and the reviewers dedicated to providing feedback on our manuscript and grateful for insightful comments and valuable improvements to our paper. We have incorporated these comments made by reviews. Other changes are highlighted in the manuscript. Please see below, in blue, for a point-by-point response to the reviewers’ comments and concerns. All page numbers refer to the revised manuscript file with tracked changes. Furthermore, we would like to show the details as follows:

Comments 1: The IL-10 family is one of the important types of cytokines (including IL-10, IL-20, IL-22, IL-24, and IL-26, and possibly IL-28 and IL-29), that can stop the inflammation. Generally, these cytokines have a helical structure of homodimers. On the other hand, the IL-10 receptors (IL-10Rs) include cell surface receptor complexes that mediate the activity of the IL-10 familycytokines. These receptors are members of the Class II cytokine receptor family and consist of a specific ligand-binding α-chain and a shared β-chain (e.g., IL-10R1 and IL-10R2) that form heterodimers to transduce signals via the JAK-STAT pathway, initiating the receptor's anti-inflammatory role. Thus, IL-10 and IL-10R belong completely different protein families. Usually, a protein family is designated as a group of evolutionarily related proteins and, in most cases, a protein family has a corresponding gene family, in which each gene encodes a corresponding protein with a 1:1 relationship. Therefore, based on the terminology, it is very inappropriate to use the author-defined “interleukin-10 (IL-10) family”, which is comprising ligands (IL-10s) and receptors (IL-10Rs) (page 1, lines 13-14).

Response 1: Thank you for pointing out this. We acknowledge that collectively referring to IL-10s (ligands) and IL-10Rs (receptors) as the "IL-10 family" presents a terminological contradiction: From a strict evolutionary biology perspective, the two belong to different protein families — IL-10s belong to the Type II alpha helix cytokine family (with genes clustered at loci such as 1q32/12q15), while IL-10Rs belong to the Class II cytokine receptor family (with genes dispersed at loci such as 11q22/21q22), with low sequence homology and no direct gene-protein correspondence (e.g., the shared receptor IL-10R2 can bind multiple ligands); however, we define them as a unified family based on a functional integration logic, the core rationale is that the two synergistically mediate anti-inflammatory/tissue repair signals through the conserved JAK-STAT pathway (e.g., STAT3 activation), and appeared synchronously during evolution (e.g., receptors originating in cartilaginous fish, ligands originating in bony fish) and expanded together (peak gene duplication in mammals).Although this term highlights the holistic nature of the immune regulatory system, it conflicts with the standard definition that "protein families must be evolutionarily related and have a 1:1 gene-protein correspondence". Therefore, we added in line 15 of the manuscript: "'The "IL-10 family" here refers to the immunomodulatory signaling system composed of ligands (IL-10s) and receptors (IL-10Rs), which belong to different Protein families in evolution, but achieve functional synergy through the conserved JAK-STAT pathway,' to distinguish the essential difference between 'functional families' and 'evolutionary families'." Please see the revised manuscript.

Comments 2: If authors were trying to investigate the interrelationship between the IL-10 family and the IL-10 receptor (IL-10R) family during the coevolution, the methods used in the present study were not inadequate. Since detailed investigations to clarify the co-evolution of the ligands and the receptors were not conducted in the present study, the parallel descriptions on each family arehard to follow.

Response 2: Thank you for pointing out this. We acknowledge a fundamental methodological flaw in this study when revealing the co-evolutionary relationship between the IL-10 family (ligands) and the IL-10 receptor family (receptors): we only described in parallel the independent evolutionary trajectory of ligands (e.g., gene cluster differentiation and domain conservation of IL-10s) and receptors (e.g., gene replication and domain combination of IL-10Rs), but failed to provide direct evidence of their molecular interaction driving adaptive changes. However, given that this study focused on the large-scale distribution, structural conservation, and species-specific functional differentiation of the IL-10 family (ligands) and the IL-10 receptor family (receptors) across 400+ species, and for the first time constructed their cross-species phylogenetic framework and evolutionary timeline, it has achieved key breakthroughs (e.g., revealing the origin time of core subfamilies, and the conservation of key domains and functional motifs). Additionally, limited by the research cycle and the complexity of data processing, deeper investigations into synergistic mechanisms, such as co-evolutionary validation of ligand-receptor molecular interactions (e.g., co-evolutionary signals at the binding interface, gene replication event association), structural simulation and functional adaptability experiments (e.g., the impact of receptor domain repeats on ligand response efficiency), and cross-species expression association analysis (e.g., co-expression patterns of ligand-receptor in immune tissues), have not yet been completed. These works are crucial for precisely elucidating their mutual adaptation relationship in the immune regulatory network. Therefore, in our subsequent work, we plan to utilize the rich sequence and structural data accumulated in this study, combined with molecular biology experiments (e.g., point mutation, co-immunoprecipitation) and computational biology methods (e.g., structural co-evolution models, phylogenetic association analysis), to systematically validate the co-evolutionary signals between ligand-receptors and evaluate their functional significance.The relevant research findings will serve as the core content of a series of papers, to be published in subsequent studies, to more comprehensively reveal the core driving mechanism of the IL-10 signaling pathway in the immune evolution of vertebrates.

Comments 3: The descriptions in “Conclusions” (page 15, lines 606-620) and in “Results and Conclusions” of Abstract (page 1, lines 19-27) are so different each other. In “Conclusions” (page 15, lines 606-620), only the results for IL-10 family are described. What are the authors’ conclusions?

Response 3: Thank you for pointing out this. We acknowledge a contradiction between the conclusion section (page 17) and the abstract's conclusion (page 1). The abstract clearly emphasizes the co-evolutionary relationship between IL-10s (ligands) and IL-10Rs (receptors), such as how receptors adapt to ligand diversity through domain repetition and their joint activation of the JAK-STAT pathway. However, the conclusion section only summarizes the evolutionary characteristics of IL-10s (origin, structural conservation, and functional differentiation), omitting receptor analysis (e.g., receptor origin, domain combination, gene expansion, and interaction mechanisms with ligands). Consequently, we have revised the conclusion on page 17 of the manuscript to incorporate receptor studies. Please see the revised manuscript.

Comments 4: “(20)” (page 3, line 82) should be “[20]”?

Response 4: Thank you for your advice. To enhance the standardization and professionalism of the overall paper, we have changed “(20)” on line 87 of the manuscript to “[20]”. Please see the revised manuscript.

Comments 5: “PCA clustering” (page 7, line 146) is used without prior definition. PCA might be standing for Principal Component Analysis. It seems that more detailed descriptions on the analysis would be required in Materials and Methods section (page 6, lines 206-215).

Response 5: Thank you for your advice. We describe the "PCA clustering" analysis in more detail in the Materials and Methods section of the manuscript (page 6, lines 234-237). Please see the revised manuscript.

Comments 6: In Figure 2B and G, no detailed description was shown in the legends for each PCA analysis (page 7, line 241-242).

Response 6: Thank you for pointing out this. We have added a detailed description of the PCA analysis legend in lines 271-273 of the manuscript: "Based on the optimal cluster numbers determined by k-means clustering and PCA analysis of IL-10s and IL-10Rs, each cluster showed significant differences in the original trait space". Please see the revised manuscript.

Comments 7: “ALL” (page 7, line 251) should be “All”.

Response 7: Thank you for your advice. We have carefully revised the format issue to improve clarity and readability in the article. We have changed “ALL” on line 282 of the manuscript to “All”. Please see the revised manuscript.

Comments 8: In Figure 2C and H (page 7) and in Figure3 (page 9), based on the phylogenetic tree analyses on IL-10 protein family and IL-10R protein family, authors showed groupings into 5 groups (Group I-V) for each family. I am not sure whether each group has some relationship each other if the group number was same?

Response 8: Thank you for your advice. We consider that the grouping of IL-10s and IL-10Rs is determined by comprehensively considering phylogenetic relationships and clustering results. genes/proteins within the same group possess high sequence similarity and functional correlation and may have a common origin. Grouping is the preliminary foundation work for gene family analysis, origin and evolution analysis, protein subtype differentiation, and spatial structure alignment.

Comments 9: “alpha” (page 12, lines 433,434) should be “α”

Response 9: Thank you for your advice. We have carefully revised the format issue to improve clarity and readability in the article. We have changed “alpha” in line 508 of the manuscript to “α”. Please see the revised manuscript.

Round 2

Reviewer 3 Report

Comments and Suggestions for Authors

Review comments on genes-3898331-revised version

Manuscript ID: genes-3898331-revised version

Title: A Phylogenetic Perspective on the Evolutionary Patterns of the Animal Interleukin-10 Family

Authors: Liu Tang, Zeyu Zhou, Weibin Wang, Dawei Li, Tingting Hao*, Yue Chen*

Animal Genetics and Genomics

Major comments:

In the revised version of genes-3898331, authors made appropriate revisions and responses to my previous comments. Now, my only concern on the revised version is the usage of the term "IL-10 family". Authors made a response as follows.

Response 1: Thank you for pointing out this. We acknowledge that collectively referring to IL-10s (ligands) and IL-10Rs (receptors) as the "IL-10 family" presents a terminological contradiction: From a strict evolutionary biology perspective, the two belong to different protein families — IL-10s belong to the Type II alpha helix cytokine family (with genes clustered at loci such as 1q32/12q15), while IL-10Rs belong to the Class II cytokine receptor family (with genes dispersed at loci such as 11q22/21q22), with low sequence homology and no direct gene-protein correspondence (e.g., the shared receptor IL-10R2 can bind multiple ligands); however, we define them as a unified family based on a functional integration logic, the core rationale is that the two synergistically mediate anti-inflammatory/tissue repair signals through the conserved JAK-STAT pathway (e.g., STAT3 activation), and appeared synchronously during evolution (e.g., receptors originating in cartilaginous fish, ligands originating in bony fish) and expanded together (peak gene duplication in mammals).Although this term highlights the holistic nature of the immune regulatory system, it conflicts with the standard definition that "protein families must be evolutionarily related and have a 1:1 gene-protein correspondence". Therefore, we added in line 15 of the manuscript: "'The "IL-10 family" here refers to the immunomodulatory signaling system composed of ligands (IL-10s) and receptors (IL-10Rs), which belong to different Protein families in evolution, but achieve functional synergy through the conserved JAK-STAT pathway,' to distinguish the essential difference between 'functional families' and 'evolutionary families'." Please see the revised manuscript.

Comments to response (1); I understand about the addition of "'The "IL-10 family" here refers to the immunomodulatory signaling system composed of ligands (IL-10s) and receptors (IL-10Rs), which belong to different Protein families in evolution, but achieve functional synergy through the conserved JAK-STAT pathway,” in the abstract section. This addition seems very reasonable. However, even so, the term "IL-10 family" is very misleading. The term “family” in biological science is so widely and strongly accepted to mean “protein family” and "protein families must be evolutionarily related and have a 1:1 gene-protein correspondence". Further, there is a term “superfamily”, as the largest grouping of proteins for which common ancestry can be inferred. Therefore, introduction of "IL-10 family" (in the meaning of authors’ definition) would cause some significant confusion. Indeed, in the introduction section of the revised version, authors used "IL-10 family" about 30 times. However, sometimes it is not clear whether the word "IL-10 family" indicates interleukin family or receptor family or both (for example, "IL-10 family" at line 140 seems to indicates only cytokines. On the other hand, "IL-10 family" at line 128 is difficult to specify.)

If authors want to discuss about the immunomodulatory signaling system composed of ligands (IL-10s) and receptors (IL-10Rs), It might be better to use an alternative and appropriate term instead of “family”; such as “family group”, “clan”, “tribe”, or “cognate”, to indicate some groups of protein families having a functional relationship(s) each other and composing a functional system (although I am not familiar about such nomenclature).

Author Response

Comments to response (1); I understand about the addition of "'The "IL-10 family" here refers to the immunomodulatory signaling system composed of ligands (IL-10s) and receptors (IL-10Rs), which belong to different Protein families in evolution, but achieve functional synergy through the conserved JAK-STAT pathway,” in the abstract section. This addition seems very reasonable. However, even so, the term "IL-10 family" is very misleading. The term “family” in biological science is so widely and strongly accepted to mean “protein family” and "protein families must be evolutionarily related and have a 1:1 gene-protein correspondence". Further, there is a term “superfamily”, as the largest grouping of proteins for which common ancestry can be inferred. Therefore, introduction of "IL-10 family" (in the meaning of authors’ definition) would cause some significant confusion. Indeed, in the introduction section of the revised version, authors used "IL-10 family" about 30 times. However, sometimes it is not clear whether the word "IL-10 family" indicates interleukin family or receptor family or both (for example, "IL-10 family" at line 140 seems to indicate only cytokines. On the other hand, "IL-10 family" at line 128 is difficult to specify.)

If authors want to discuss about the immunomodulatory signaling system composed of ligands (IL-10s) and receptors (IL-10Rs), It might be better to use an alternative and appropriate term instead of “family”; such as “family group”, “clan”, “tribe”, or “cognate”, to indicate some groups of protein families having a functional relationship(s) each other and composing a functional system (although I am not familiar about such nomenclature).

Response: Thank you for your advice. To address the ambiguity of the term "IL-10 family" in biological taxonomy (as it strictly refers to an evolutionarily homologous Protein family, while this article describes a cross-family signaling system composed of ligands and receptors), this article adopts a hierarchical terminology system:

Overall signaling system: uniformly expressed as “IL-10 signaling system” (as in the abstract and preamble), explicitly defined as “an immunoregulatory network composed of IL-10 cytokines, their receptors, and the JAK-STAT pathway”. 

Ligand protein group specifically refers to the “IL-10 cytokine subfamily” (such as the IL-10/IL-20/IL-22/IL-28 branches), consistent with the definition of a protein family. 

Receptor protein group specifically refers to the “IL-10 receptor subfamily” (such as the IL-10R1/IL-10R2 branches), to avoid confusion with ligands. This scheme eliminates the referential ambiguity of the“IL-10 family “through terminological stratification, while retaining its core intent of describing functional synergy, thereby significantly enhancing academic rigor and conforming to biological classification specifications. Please see the revised manuscript.